# Integrated analysis of telomerase enzymatic activity unravels an association with cancer stemness and proliferation

Nighat Noureen[1,2], Shaofang Wu[3], Yingli Lv[1], Juechen Yang[1], W. K. Alfred Yung[3], Jonathan Gelfond[2], Xiaojing Wang[1,2], Dimpy Koul[3], Andrew Ludlow [4] & Siyuan Zheng [1,2 ✉]

Active telomerase is essential for stem cells and most cancers to maintain telomeres. The enzymatic activity of telomerase is related but not equivalent to the expression of TERT, the catalytic subunit of the complex. Here we show that telomerase enzymatic activity can be robustly estimated from the expression of a 13-gene signature. We demonstrate the validity of the expression-based approach, named EXTEND, using cell lines, cancer samples, and non-neoplastic samples. When applied to over 9,000 tumors and single cells, we find a strong correlation between telomerase activity and cancer stemness. This correlation is largely driven by a small population of proliferating cancer cells that exhibits both high telomerase activity and cancer stemness. This study establishes a computational framework for quantifying telomerase enzymatic activity and provides new insights into the relationships among telomerase, cancer proliferation, and stemness.

[1] Greehey Children's Cancer Research Institute, UT Health San Antonio, San Antonio, TX, USA. [2] Department of Population Health Sciences, UT Health San Antonio, San Antonio, TX, USA. [3] Department of Neuro-Oncology, MD Anderson Cancer Center, Houston, TX, USA. [4] Department of Movement Science, University of Michigan, Ann Arbor, MI, USA. ✉email: zhengs3@uthscsa.edu

Telomerase is the ribonucleoprotein complex that adds telomeric repeats to telomeres at chromosome ends. In the absence of telomerase, telomeres progressively shorten due to incomplete replication of chromosome ends[1]. Persistent telomere shortening leads to senescence and crisis, thus continuously dividing cells including stem cells and most cancer cells require active telomerase to maintain telomere lengths[2]. Loss of telomerase activity results in degenerative defects and premature aging[3,4]. In contrast, reactivation of telomerase enables malignant transformation and cancer cell immortality[5,6]. These observations emphasize the pivotal role of telomerase in many human health concerns and highlight the need for close monitoring of its activity.

The telomerase complex is composed of the reverse transcriptase subunit TERT, the template containing non-coding RNA TERC, and accessory proteins such as dyskerin (DKC1) and telomerase Cajal body protein 1 (TCAB1). The core subunits are the catalytic subunit TERT and the RNA template TERC. In vivo, the processive extension of telomeres, that is, telomerase processivity, requires binding of telomerase to telomeres through a six-protein complex, shelterin, specifically its component protein TPP1[7,8]. Another shelterin protein POT1 bridges TPP1 and chromosome 3′ overhang, the DNA substrate of telomerase.

While TERC is thought to be abundant and ubiquitously expressed[9,10], TERT is transcriptionally repressed in most somatic cells[11]. Some cancer lineages (e.g., brain, liver, skin, and bladder) frequently acquire recurrent mutations in the TERT promoter (TERTp) region, predominantly at −124 and −146 loci upstream from TERT transcription start site[12–15]. These C > T mutations create consensus binding sites for GABP transcription factors, alter chromatin states, and enhance the transcriptional output of TERT[16–20]. In bladder cancer, the promoter mutations correlate with increased TERT expression and telomerase enzymatic activity[21].

Enzymatic activity is a fundamental metric of telomerase. Several protocols have been established to measure telomerase enzymatic activity, including the polymerase chain reaction (PCR)-based telomeric repeat amplification protocol (TRAP) assay and direct enzymatic assays[22–24]. These assays allowed for investigations of associations between telomerase activity and clinical and histopathologic variables in cancer and other diseases[25,26], and regulation of telomerase activity by each component of the telomerase complex. Ectopic expression of TERT and TERC, in many cases TERT alone, increases telomerase enzymatic activity[27–29]. These data led to the view that TERT is the limiting component for telomerase activity.

However, emerging data challenge the use of TERT expression as a surrogate for telomerase enzymatic activity. First, the TERT gene can transcribe >20 splicing isoforms[30], but only the full-length transcript bearing all 16 exons can produce the catalytic subunit[31–33]. Second, single-cell imaging studies showed that most TERT mRNAs localize in the nucleus, but not in the cytoplasm, and thus are not translated[34]. Third, endogenous TERT protein and TERC are far more abundant than the assembled telomerase complex in cancer cell lines[23]. Finally, TERC and accessory proteins can also impact telomerase activity. For example, telomerase activity in human T cells has been reported to relate to TERC levels rather than TERT[35]. In addition, mutations in DKC1 and TERC both cause dyskeratosis congenita, a rare genetic syndrome related to impaired telomerase[10,36].

We and others previously analyzed telomere lengths, telomere maintenance mechanisms (TMMs), and TERT expression in cancer[37–39]. However, telomerase enzymatic activity has not been systematically characterized largely due to the lack of a rigid, scalable quantification method. In this work, we report a systematic analysis of telomerase activity in cancer. This analysis was enabled by a telomerase activity prediction algorithm presented here and made available for the wider community, EXpression-based Telomerase ENzymatic activity Detection (EXTEND).

## Results

**Rationale and overview of EXTEND.** An overview of the EXTEND algorithm is shown in Supplementary Fig. 1. We posited that comparing the expression of telomerase-positive and -negative tumors could yield a telomerase activity signature. Most epithelial tumors express TERT and use telomerase to maintain telomeres. Without experimental evidence, however, TERT expression cannot reliably indicate positive telomerase for the reasons noted above. We instead used TERTp mutation to stratify such tumors, reasoning that the presence of this genetic change likely reflects evolutionary selection for telomerase as the predominant TMM. We further assumed that tumors with alternative lengthening of telomere (ALT) phenotype, an alternate TMM mechanism, were negative controls. The ALT phenotype is usually determined through ALT-associated promyelocytic leukemia nuclear bodies, extrachromosomal telomeric DNA C-Circles, or ALT-associated telomere foci[40,41]. Large cohorts of experimentally confirmed ALT tumors are not yet publicly available, largely due to the rarity of the phenotype. However, mutations in ATRX and its interacting partner DAXX are nearly perfectly correlated with ALT[42]. We thus searched the TCGA dataset for cancer types with both high frequencies of mutations in the TERTp and ATRX and DAXX. This search identified lower-grade glioma (LGG) (Supplementary Fig. 2), a cancer type demonstrating strong mutual exclusivity of the two TMMs[43].

We identified 108 TERT co-expressing genes in the LGG dataset. These genes were further intersected with genes upregulated in the TERTp mutant tumors. The resulting 12 genes were complemented with TERC, the RNA subunit of the telomerase complex, giving rise to a 13-gene signature. Seven of the 13 genes were highly expressed in testis, but low in other tissues. None of the signature genes except TERT was cataloged by the expert-curated Cancer Gene Census[44] (as of July 2020), although LIN9 and HELLS were recently implicated in cancer[45,46]. Mutations in HELLS, a gene encoding a lymphoid-specific helicase, cause the centromeric instability and facial anomalies (ICF) syndrome, a genetic disorder associated with short telomeres[47]. Another signature gene POLE2 was a subunit of DNA polymerase epsilon, a complex previously linked to telomerase c-strand synthesis[48]. A summary of the signature genes, including their tissue expression pattern, function, and expression pattern in LGG, was provided in Supplementary Data 1. Pathway enrichment analysis suggested an overrepresentation of the signature genes in the cell cycle (false discovery rate (FDR) = 1.95e − 4), particularly S phase (FDR = 0.01, Supplementary Data 2), a narrow time window when telomerase is active in extending telomeres[49]. To examine if this signature was LGG specific, we tested the correlation between the expression of signature genes (excluding TERT and TERC) and TERT in 32 cancer types. We observed positive correlation in 63% of all gene–cancer type pairs, and all genes were positively correlated with TERT across pan-cancer, suggesting that this signature is not LGG specific (Supplementary Fig. 3).

To score this signature, we designed an iterative rank-sum method (Supplementary Fig. 1). This method first divides the signature into a constituent component (TERT and TERC) and a marker component (the other 11 genes). The constituent component is scored by the maximum ranking of TERT and TERC, whereas the marker component is scored by the rank sum of the signature genes. Because the size of the constituent component is much smaller than the marker component, we

adjusted its contribution to the final score by a factor determined based on the correlation between the constituent component score and the marker component score. Using cancer cell lines from the Cancer Cell Line Encyclopedia (CCLE), we estimated that the constituent component generally contributes <20% to the final scores (Supplementary Fig. 4).

Since *TERC* lacks a long poly(A) tail, its measurement is less robust with a poly(A)-enriched mRNA-sequencing protocol. Thus, we tested the stability of EXTEND across sequencing protocols. Using samples sequenced by both ribosomal RNA depletion and poly(A) enrichment protocols[50], we found that *TERC* was indeed less concordant than *TERT* between the two protocols (Rho = 0.71 vs. 0.98), but EXTEND scores nevertheless agreed well (Rho = 0.96, $P = 1.9e − 6$; Supplementary Fig. 5).

**Validation and comparison with *TERT* expression in cancer.** We first validated EXTEND with cancer cell lines. We performed semi-quantitative TRAP assays on 28 patient-derived glioma sphere-forming cells (GSCs) (see "Methods"). EXTEND scores were significantly correlated with experimental readouts (Rho = 0.48, $P = 0.01$; Fig. 1a). Although this correlation was comparable to *TERT* expression (Rho = 0.5, $P = 0.01$), EXTEND demonstrated superior performance in differentiating two recently determined ALT lines, GSC5–22 and GSC8–18[40] (Supplementary Fig. 6). Using 11 BLCA lines with available RNA-sequencing (RNAseq) data, we compared EXTEND predictions with results from direct enzymatic assays[21]. EXTEND scores and cell line telomerase activity were significantly correlated (Rho = 0.72, $P = 0.01$), whereas the correlation for *TERT* did not reach statistical significance (Rho = 0.55, $P = 0.08$) (Fig. 1b and Supplementary Fig. 7). We then measured the telomerase activity of 15 lung cancer cell lines using digital droplet TRAP assays[24]. EXTEND outperformed *TERT* expression in predicting telomerase on these cell lines (Rho = 0.65, $P = 0.01$ vs. Rho = 0.48, $P = 0.07$) (Fig. 1c and Supplementary Fig. 8).

We next tested EXTEND in two cancer types enriched with ALTs[51], liposarcoma, and neuroblastoma. In liposarcoma[52], telomerase-positive tumors had significantly higher EXTEND scores than ALT tumors ($P < 0.01$, Student t test; Fig. 1d). EXTEND outperformed *TERT* expression in differentiating ALTs and telomerase-positive cases in this dataset ($P = 7e − 06$ vs. 0.003 for cell lines and $P = 0.001$ vs. 0.0091 for tumors; two-sided Student's *t* test; Supplementary Fig. 9). Neuroblastomas were previously divided into five groups based on TMMs: TERTp rearrangement, *MYCN* amplification, *TERT* expression high, ALT, and no TMM[41]. Enzymatic assays suggested that the first three groups were telomerase positive, whereas the latter two had very low telomerase activity[41]. Consistent with these observations, EXTEND estimated higher telomerase activity in the three telomerase-positive groups (Fig. 1e). The *TERT* high group was estimated to have relatively lower scores than the other two telomerase-positive groups, a pattern also consistent with the experimental data[41] (Supplementary Fig. 10).

**Telomerase activity in non-neoplastic and embryonic samples.** We then applied EXTEND to tissue samples from the Genotype Tissue Expression (GTEx) dataset (Supplementary Data 3). These samples were collected post mortem from healthy donors, except for a few transformed cell lines. Among the 52 sub-tissue types, we observed the highest EXTEND scores in Epstein–Barr virus-transformed lymphocytes, testis, transformed skin fibroblasts, and esophageal mucosa, a tissue with a high self-renewal rate (Fig. 2a and Supplementary Fig. 11). The testis was recently shown to have the longest average telomere lengths among human tissues[53]. Highly differentiated tissue

skeletal muscle samples had the lowest average scores. Skin-transformed fibroblasts had almost negligible *TERT* expression despite a relatively high EXTEND score. In contrast, brain tissues, including substantia nigra (brain_SN), putamen, nucleus accumbens (brain_NA), and caudate, had low EXTEND scores, but expressed *TERT* at a detectable level. To explain this disparity, we examined *TERT* alternative splicing. Compared with the testis, most expressed *TERT* were short-spliced forms in the brain and thus could not encode catalytically active proteins (Supplementary Fig. 12).

Next, we analyzed human tissues during embryonic development. We focused on the heart and liver, two organs with distinct self-renewal behaviors. Cardiomyocytes were thought to lose their proliferative ability after birth[54], whereas telomerase-expressing hepatocytes robustly repopulate the liver in homeostasis throughout adulthood[55,56]. Using a recently published dataset[57], we analyzed both organs through embryonic weeks 4–20 and after birth. For heart, EXTEND scores dropped considerably between the 12th and 13th embryonic weeks, but remained at modest levels in infants and toddlers (Fig. 2b). A further drop was observed when entering adulthood. These observations agreed with previous studies that reported decreased telomerase activity after the 12th embryonic week in the heart tissue[58], and that infant hearts have higher telomerase activity than those of adults[59].

We did not observe significant decreases in EXTEND scores for liver during late embryonic weeks, despite the declining *TERT* expression after around 10 weeks. EXTEND scores largely remained stable across the lifespan, although their levels were much lower than in fetal samples (Fig. 2c, $P = 1.1e − 9$).

Finally, we applied EXTEND to samples from a patient with dyskeratosis congenita[60]. This patient carried loss-of-function mutations in poly(A)-specific ribonuclease (*PARN*), a gene required for *TERC* maturation. Inhibition of poly(A) polymerase PAP-associated domain–containing 5 (*PAPD5*) counteracts *PARN* mutations to increase *TERC* levels and telomerase activity[60]. EXTEND predicted higher telomerase activity in *PARN* mutant cells compared with wild-type controls upon *PAPD5* knockdown, confirming the earlier finding (Fig. 2d).

**Landscape of telomerase activity in cancer.** We next analyzed >9000 tumor samples and 700 normal samples from TCGA (Supplementary Data 4). We first evaluated if EXTEND scores were also reflective of telomerase processivity in cancer. POT1 is a shelterin component that regulates telomerase processivity through interactions with chromosome 3′ overhang and TPP1, the telomerase anchor to telomeres[61]. TPP1-POT1 heterodimer has been reported to enhance telomerase processivity[62–64]. EXTEND scores were positively correlated with *POT1* expression across pan-cancer ($Rho = 0.22, P < 2.2e − 16$). No correlation was observed between *TERT* and *POT1* ($Rho = −0.004, P = 0.71$) (Supplementary Fig. 13). We did not find significant associations between telomere length and EXTEND scores in TCGA cohorts after adjusting for multiple hypothesis testing (Supplementary Fig. 14).

As expected, tumors had significantly higher EXTEND scores than matched normal samples (Fig. 3a; $P < 2.2e − 16$, t test). Similar to the GTEx sample analyses, scores varied across normal tissues. Gastrointestinal organs (esophageal—ESCA (esophageal carcinoma), stomach—STAD (stomach adenocarcinoma), colorectal—CRC (colorectal adenocarcinoma)), mammary gland (breast—BRCA (breast carcinoma)), and reproductive organs (uterus endometrial—UCEC (uterine corpus endometrial carcinoma)) had overall higher scores. EXTEND scores varied across cancer types, with kidney (KIRP (kidney renal papillary

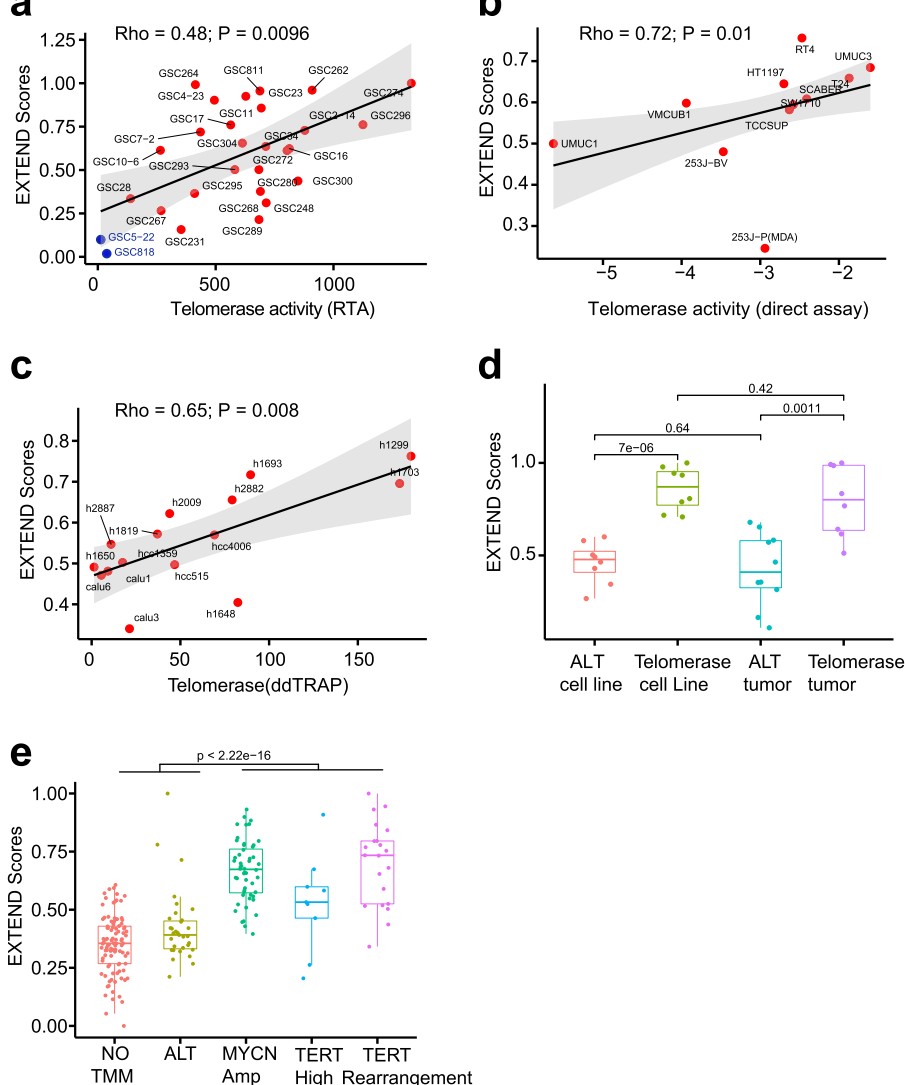

**Fig. 1 EXTEND Validation. a** Correlation between EXTEND score and TRAP assay readouts in 28 glioma sphere-forming cell lines. Two ALT cell lines are labeled in blue. For (**a**–**c**), Spearman's correlation was used to calculate $P$ value and Rho, and shade indicates 95% CI of the regression calculated by ggplot2 using default parameters. **b** Correlation between EXTEND score and direct enzymatic assay results in 11 bladder cancer cell lines. **c** Correlation between EXTEND score and digital droplet TRAP assay in 15 lung cancer cell lines. **d** EXTEND scores across ALT and telomerase-positive tumors in liposarcomas ("GSE14533"). Both telomerase-positive tumors ($n = 8$) and cell lines ($n = 8$) show significantly higher EXTEND scores than ALT samples (cell lines, $n = 8$ and tumors, $n = 10$). **e** EXTEND scores across the five TMM groups of neuroblastoma ("GSE120572"). Telomerase-positive groups (*MYCN* amplification ($n = 52$), *TERT* high ($n = 9$), and *TERT* rearrangement ($n = 21$)) show significantly higher scores than ALT ($n = 31$) and no TMM ($n = 99$) groups. Boxplots, interquartile ranges (25–75th percentile); middle bar defines median and the minima and maxima are within 1.5 times the interquartile range of the lower and higher quartile. Statistical differences were assessed using two-sided Student's $t$ test in (**d**) and (**e**). Data used are available in Source Data.

carcinoma), KIRC (kidney renal clear carcinoma)), thyroid (THCA (thyroid carcinoma)), prostate (PRAD (prostate adenocarcinoma)), and pancreatic (PAAD (pancreatic adenocarcinoma)) demonstrating the lowest scores (Fig. 3a). These cancer types also appeared to have the smallest differences between normal and cancer samples. The small difference for pancreatic cancer may reflect this cancer type's high impurity.

Using seven cancer types where each of the four stages had at least ten samples, we found in four of seven tested cancer types (THCA, KIRC, KIRP, and LUAD (lung adenocarcinoma)), EXTEND scores increased in high-stage tumors, suggesting higher telomerase activity in advanced-stage tumors consistent with previous reports[65–68]. In contrast, STAD and CRC showed the highest scores in stage I tumors (Fig. 3b). In melanoma,

metastatic cancers exhibited higher scores than primary and regionally invasive tumors (Fig. 3c).

Given the correlation between EXTEND score and tumor stage, we further tested its association with tumor molecular subtypes. EXTEND scores were significantly different in at least one subtype classification in all tested cancer types (Supplementary Fig. 15), suggesting that telomerase activity may be an important factor underlying the molecular heterogeneity of cancer.

We also examined associations between telomerase activity and patient prognosis. We categorized EXTEND scores into low and high tumor groups based on the median and performed univariate and multivariate survival analyses. Five cancer types (adrenocortical carcinoma (ACC), kidney chromophobe (KICH),

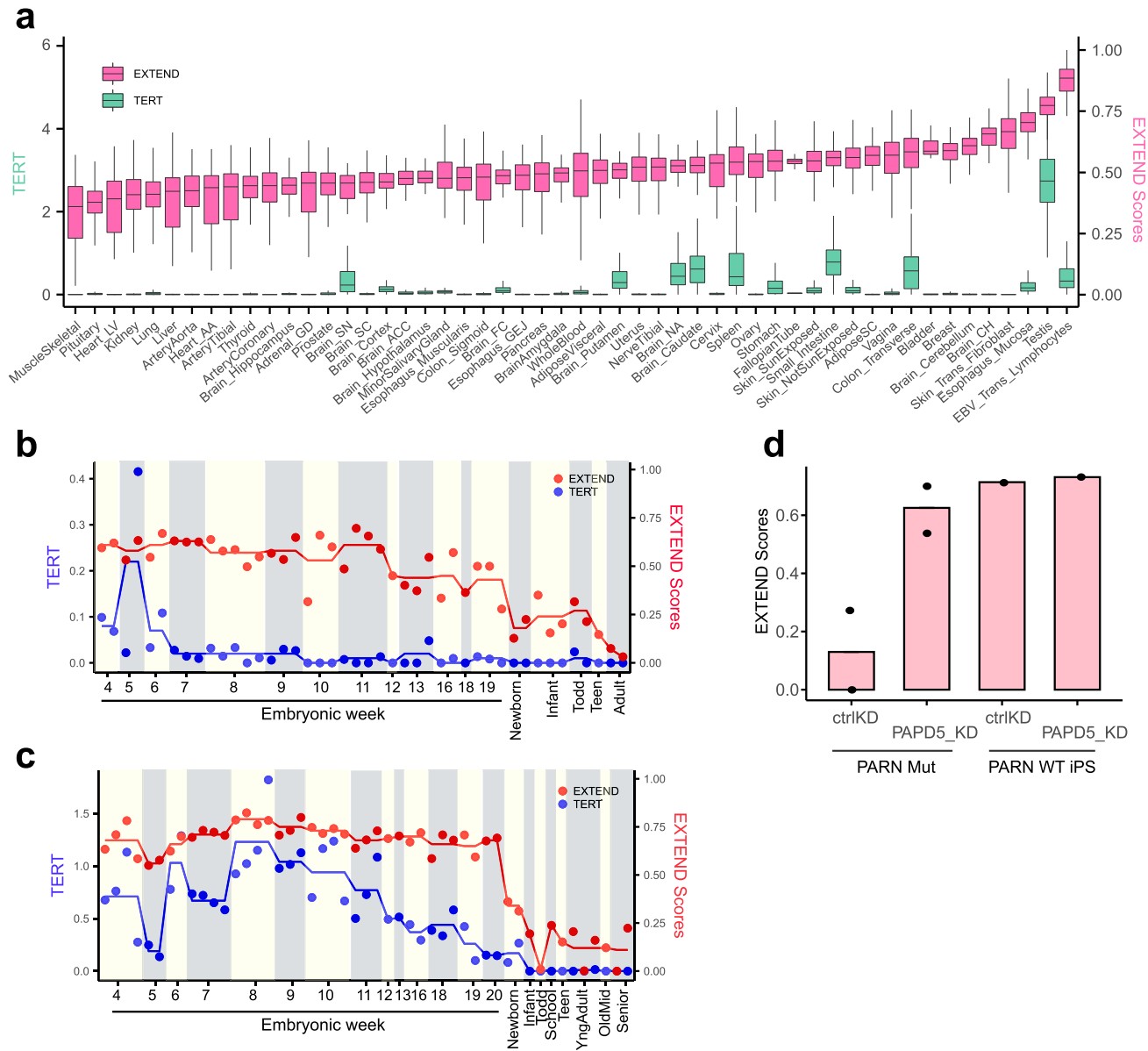

**Fig. 2 EXTEND scores across normal tissues and during embryonic development. a** EXTEND scores and *TERT* expression across 52 sub-tissues across Genotype Tissue Expression (GTEx) data ($n = 11,688$). The left *y*-axis indicates *TERT* expression (sea green), while the right *y*-axis represents EXTEND scores (pink). Boxplot interquartile ranges (25–75th percentile); middle bar defines median and the minima and maxima are within 1.5 times the interquartile range of the lower and higher quartile. **b, c** *TERT* expression (blue) and EXTEND scores (red) across tissue development phases in (**b**) heart tissue and (**c**) liver tissue. Red and blue lines represent the mean of EXTEND scores and *TERT* expression of samples from each age group. **d** EXTEND predicts higher telomerase activity in *PARN* mutant ($n = 2$) and *PARD5* knockout ($n = 2$) sample (left two bars), whereas no such effect is observed in *PARN* wild-type iPS cells ($n = 1$ for each case) (right two bars). Data are downloaded from "GSE81507." Data used for the figure is available in Source Data.

KIRP, LUAD, and SARC (sarcoma)) exhibited worse overall survival in the high score group, whereas STAD and thymoma (THYM) exhibited the opposite pattern (Fig. 3d). However, only ACC and STAD remained significant in the multivariate analysis when controlling for tumor stage and patient age at diagnosis, likely due to the association between telomerase activity and tumor stage.

Next, we correlated EXTEND scores with the ten oncogenic signaling pathways recently curated by Pan-Cancer Atlas to identify potential regulators of telomerase[69]. Cell cycle, p53, Myc, and receptor tyrosine kinase (RTK) pathways were largely positively correlated with EXTEND scores, whereas the tumor growth factor-beta (TGF-beta) and Wnt pathways were negatively correlated (Fig. 3e). The positive correlation between cell

cycle genes and EXTEND corroborates the observation that telomere extension by telomerase occurs during cell cycle[49,70]. Myc, Wnt, and TGF-beta pathways, specifically c-myc, beta-catenin, and *TGFBR2*, directly regulate telomerase[71–75]. The expression of *PDGFRA*, a marker of mesenchymal cells, was negatively correlated with EXTEND scores in 21 of 31 cancer types. This was in stark contrast to other genes of the RTK pathway, suggesting a possible suppression of telomerase activity in cells of mesenchymal origins.

**Correlation between telomerase activity and cancer stemness.**
Increasing evidence suggests that cancers exhibit stem cell-like characteristics, although it is still controversial if cancer stemness reflects the presence of cancer stem cells or stem cell-associated

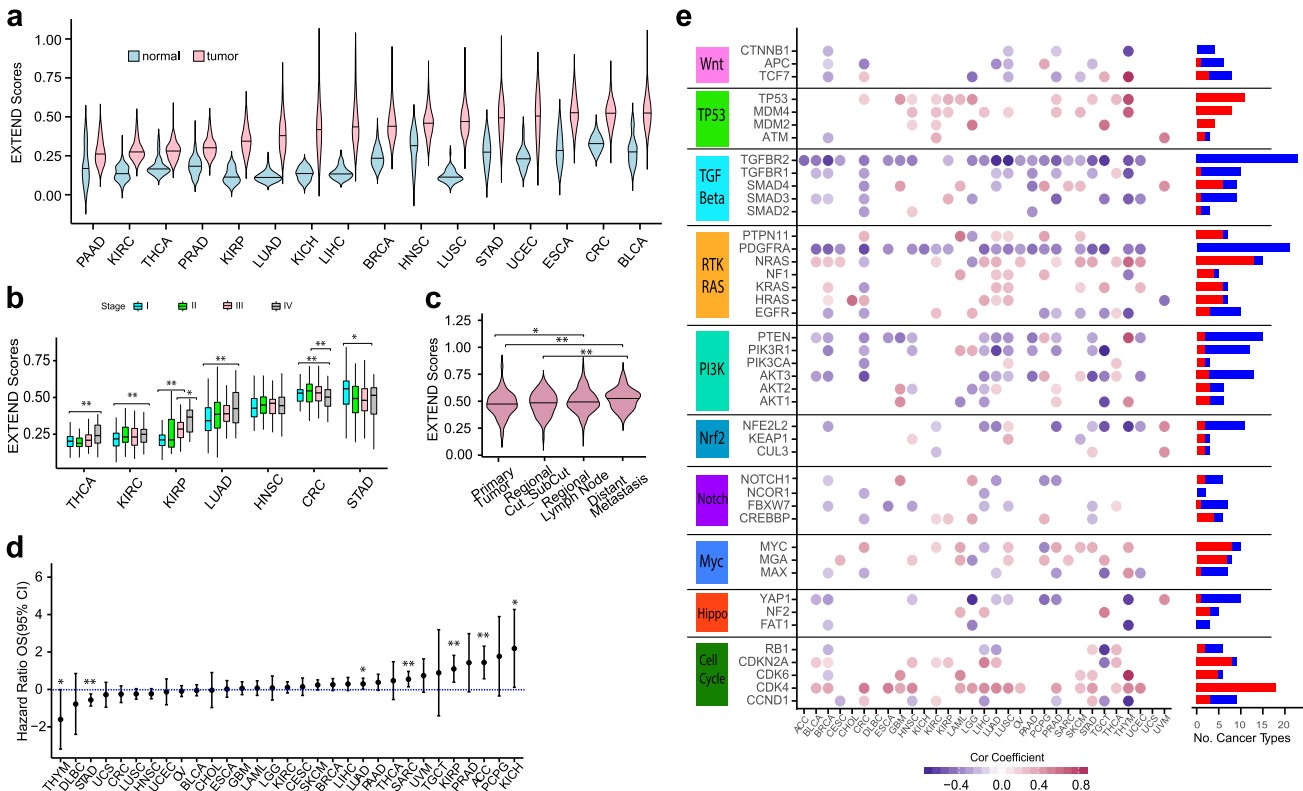

**Fig. 3 EXTEND scores in cancer. a** EXTEND scores for *TERT*-expressing tumors and normal samples across 16 TCGA cancer cohorts. Violin width indicates data point density. Middle bar, median. **b** EXTEND scores across tumor stages (CRC = 338, HNSC = 420, KIRC = 375, KIRP = 254, LUAD = 497, STAD = 374, and THCA = 499 cases in total). Only tumor types with a minimum of ten cases for each stage were used in this analysis. *P* values were calculated using two-sided *t* test. *\*P* < 0.05; *\*\*P* < 0.01. Box indicates 25–75th quantile range. Middle bar indicates median. The minima and maxima are within 1.5 times the interquartile range of the lower and higher quartile. **c** EXTEND scores of cutaneous melanomas (TCGA SKCM). Violin width indicates data point density. Middle bar, median; two-sided *t* test. *\*P* < 0.05; *\*\*P* < 0.01. **d** Hazard ratio plot based on univariate Cox regression model for 31 cancer types. Vertical bars indicate upper and lower limits of the 95% confidence interval of the hazard ratio estimates. *P* values are highlighted for significant cases only (*\*P* < 0.05; *\*\*P* < 0.01). The *y*-axis is in the natural log scale. **e** Correlations between EXTEND scores and gene expression of ten oncogenic signaling pathways across 31 cancer types. Significant correlations (FDR < 0.05) are shown in either red (positive correlation) or blue circles (negative correlation). The frequency of each gene's positive and negative correlation patterns across different cancer types is summarized on the right. Data used for the figure is available in Source Data.

programs[76]. Since active telomerase is a feature of stem cells, we examined the association between cancer stemness and telomerase activity.

We obtained stemness measurements of TCGA tumors from a recent pan-cancer analysis[77]. This cancer stemness index was calculated from an expression signature of 12,945 genes independently derived by comparing embryonic stem cells and differentiated progenitor cells. We found cancer stemness and EXTEND scores were highly correlated at the cancer-type level (Rho = 0.85, *P* = 2e − 9) (Fig. 4a). This significant correlation remained within each cancer type (FDR < 0.05), suggesting that it was cancer lineage independent. Nine of the 13 EXTEND signature genes overlapped with the stemness signature. However, removing these nine genes from the stemness signature virtually had no impact on the resulting stemness scores as they only constituted <0.1% of the stemness signature. To further validate the correlation between stemness scores and EXTEND, we performed a permutation analysis by randomly shuffling gene labels of the expression data. We then calculated empirical *P* values by comparing observed correlations with those generated from the random permutation; 27 of 31 cancer types remained highly significant (empirical *P* values <0.05) (Supplementary Data 5).

We next asked if these tumor and tissue level correlations reflected tumor cell behavior. We calculated cancer stemness

using single-cell RNAseq data from glioblastoma (GBM)[78], head and neck cancer[79], and medulloblastoma samples[80]. Although none of the datasets had the sensitivity to detect *TERT* or *TERC* expression due to the low input materials from single cells, EXTEND successfully estimated telomerase activity scores for each cell. We again observed strong positive correlations between single-cell EXTEND scores and cancer stemness (*P* < 2.2e − 16) (Figs. 4b, c and Supplementary Fig. 16a), suggesting inherent associations between the two concepts in cancer cells.

To characterize these high stemness, high telomerase cells, we identified differentially expressed genes and their corresponding pathways in these cells (Supplementary Data 6 and Supplementary Fig. 17). Cell cycle, DNA replication, and repair pathways were significantly enriched among all three cancer types (Fig. 4d), suggesting that these were cycling cells. To further verify this finding, we divided cells into G1_S, G2_M, and non-cycling based on recently published markers[78]. Cells in the G1_S phase exhibited significantly higher EXTEND scores than G2_M cells and non-cycling cells (Figs. 4e, f and Supplementary Fig. 16b). Taken together, these data support a model that a group of high stemness, high telomerase cells drive tumor growth.

This model also predicts that telomerase activity is not only a marker for tumor stemness but also a marker for tumor proliferation. Although previously documented[81,82], the relationship between telomerase activity and tumor proliferation has not

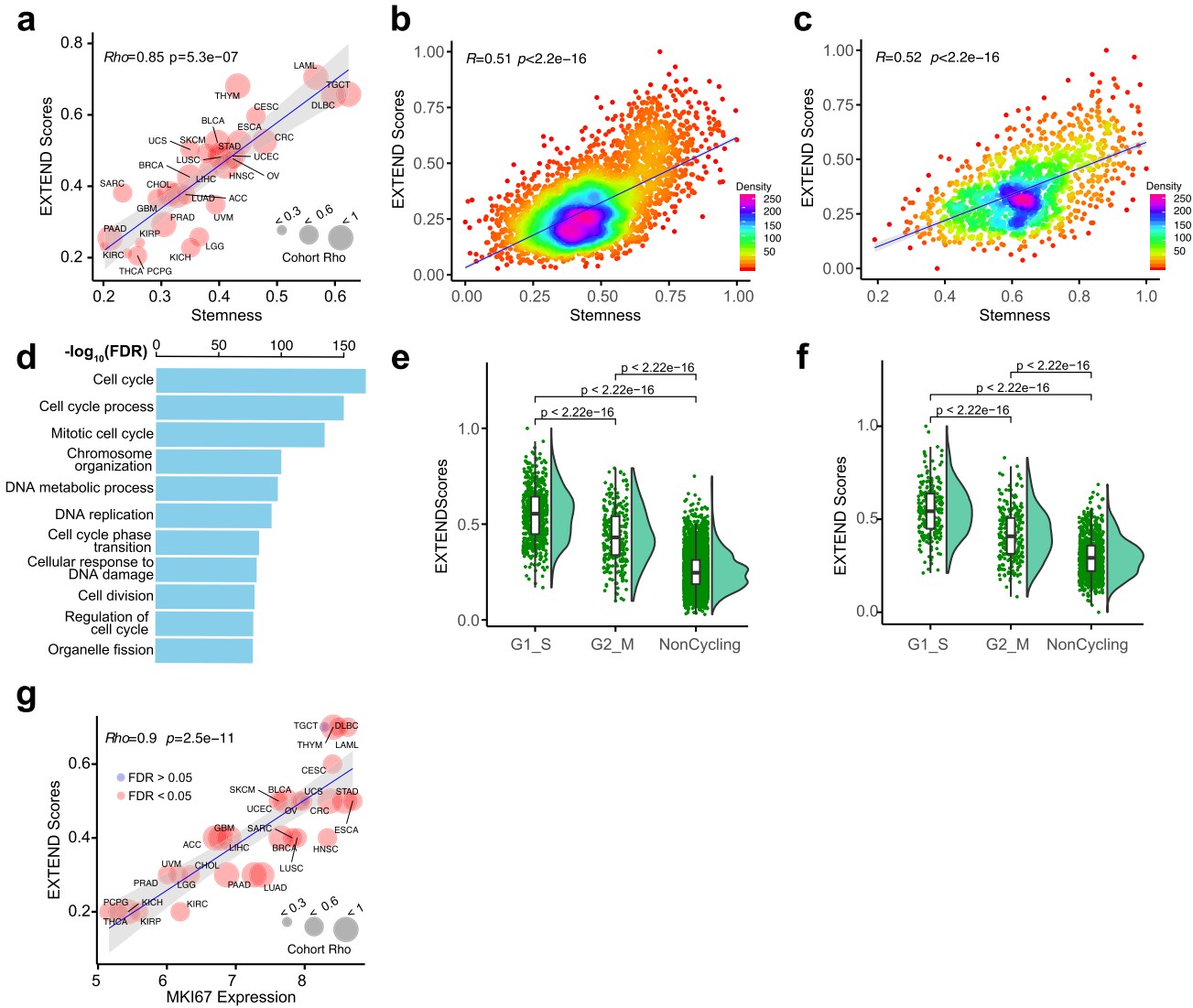

**Fig. 4 Association between EXTEND score, cancer stemness, and proliferation. a** Correlation between telomerase activity and cancer stemness across TCGA cohorts. Node size is proportional to correlation coefficients. In all cancer types, tumor stemness and telomerase activity is significantly correlated (FDR < 0.05). Shade indicates 95% CI of the regression. Spearman's correlation was used to calculate Rho and P values for all plots unless otherwise stated. **b, c** Correlation between EXTEND score and cancer stemness at single-cell level in (**b**) glioblastoma and (**c**) head and neck cancer. Each dot represents one cell. **d** Top ten pathways enriched in the high stemness, high telomerase cells from head and neck cancer. P values are calculated using two-sided t test. Box indicates 25–75th quantile range. Middle bar indicates median. The minima and maxima are within 1.5 times the interquartile range of the lower and higher quartile. **e, f** EXTEND scores of cycling cells (G1–S phase and G2–M phase) and non-cycling cells in (**e**) glioblastoma and (**f**) head and neck cancer. **g** Correlation between EXTEND scores and proliferation marker *MKI67* expression across 31 cancer types. This correlation is significant in all but two cancer types (FDR < 0.05 in blue). Shade indicates 95% CI of the regression similar to (**a**).

been quantitatively measured. Using Ki-67, a cell proliferation marker, we observed a linear positive correlation between tumor proliferation and EXTEND score on cancer type level (Rho = 0.9, $P = 2.5e - 11$) and in most tumor cohorts (FDR < 0.05). Exceptions were two tumor types that originate from reproductive organs, uterine carcinosarcoma (UCS) and testicular germ cell tumors (TGCTs) (Fig. 4g).

## Discussion

*TERT* has long been recognized as a pivotal determinant of telomerase enzymatic activity. However, it has been increasingly clear that *TERT* is involved in telomere-independent functions and its expression has limitations in predicting telomerase activity[23,32–34]. Furthermore, *TERT* has a high GC content (58% vs. genome-wide average 41% in human), making it hard to

capture in sequencing even for bulk samples. This issue is exacerbated by the low expression of *TERT*, which was estimated to be between 1 and 40 copies per cell in cancer cells[34]. Indeed, only 73% of TCGA tumors detected *TERT* expression[37], a fraction significantly lower than the TRAP assay-based estimate of telomerase-positive tumors (~90%)[22]. In single-cell RNAseq studies, it is typical that none of *TERT* reads are detected in any cells. Thus, *TERT* expression is inadequate, both biologically and technically, for guiding a systematic analysis of telomerase activity.

We here develop EXTEND to predict telomerase activity based on gene expression data. Although initially identified in brain tumors, the robustness of the signature is supported by the sustained correlation between signature genes and *TERT* in most cancer types from TCGA, except a few indolent types such as cholangiocarcinoma (CHOL), KICH, and pheochromocytoma

and paraganglioma (PCPG). Thus, caution should be exercised when applying EXTEND to such cancer types.

Using cancer cell lines of various tissues of origin, we show that EXTEND outperformed *TERT* expression in predicting telomerase activity. Analyses of GTEx and embryonic samples further supported this conclusion. Interestingly, brain tissues exhibited detectable *TERT* expression in the GTEx data but low EXTEND scores, likely due to *TERT* alternative splicing leading to isoforms that do not encode catalytically active proteins. The brain was reported to have an extreme transcriptome diversity due to alternative splicing compared with other tissues[83], thus the excessive alternative splicing of *TERT* is not surprising and unlikely specific to *TERT*. Moreover, EXTEND accurately demonstrated the point where telomerase activity is diminished during fetal heart development. These data substantiate EXTEND's validity and distinguish it from *TERT*. An important reason for EXTEND's robust performance is the 11 marker genes that have no reported functional associations with the telomerase. These genes not only reduce the impact of *TERT* expression on the final scores but also provide an avenue for estimating telomerase activity even when *TERT* or *TERC* are beyond detection sensitivity in bulk or single-cell samples.

Applying EXTEND to >9000 tumors and 700 normal samples confirmed many previously known associations but also revealed important insights. Telomerase activity, as quantified by EXTEND, is continuous rather than dichotomous. Variations in telomerase activity across cancer types can be partially explained by *TERT* expression (Rho = 0.54, *P* = 0.003). Strikingly, cancer stemness, a measure reflecting the overall similarity in self-renewal between cancer cells and embryonic stem cells, correlates significantly better than *TERT* expression with telomerase activity (Rho = 0.85, *P* = 2e − 9). This correlation coefficient indicates 72% of the variation in telomerase activity across cancer types can be explained by their differences in cancer stemness. Single-cell analysis confirmed the tight correlation between active telomerase and cancer stemness program, and further revealed that the high stemness, high telomerase cells were cycling cells. Future studies will be needed to elucidate the identities of these cells, and whether inhibiting their telomerase activity, or targeting the stemness program, or both can effectively eliminate these cells.

The lack of correlation between telomerase activity and telomere lengths was surprising but could be explained by several reasons. First of all, telomere lengths are determined by the counteracting effects of attrition during cell division and extension by telomerase. Second, telomere lengths used in our analysis were estimated based on abundances of telomeric repeats from sequencing data[37]. At best, short-read sequencing data can only estimate the average telomere length of a sample. However, telomerase preferentially acts on the shortest telomeres[5]. Furthermore, a recent study suggests that telomeric repeats can frequently insert in non-telomeric regions of the genome[38]. Thus, overall telomeric repeat counts may not precisely measure a tumor's telomere lengths.

In summary, our study demonstrates the feasibility of digitalizing telomerase enzymatic activity, a pathway fundamental to the cancer cell and stem cell survival. Our analysis establishes quantitative associations among telomerase activity, cancer stemness, and proliferation.

## Methods

**Training dataset**. EXTEND is a tool devised to predict telomerase activities using gene expression data. Training dataset utilized for EXTEND was downloaded from The Cancer Genome Atlas (TCGA) Pan-Cancer Atlas (File available on synapse. Synapse ID: syn4874822. File name: unc.edu_PANCAN_IlluminaHiSeq_RNA-SeqV2.geneExp_whitelisted.tsv). We used the TCGA LGG cohort consisting of 81 TERTp mutant samples and 123 *ATRX* altered cases (based on mutations, deletions, and structural events)[37]. No human subjects were involved in this study.

**Signature identification**. EXTEND schema is composed of two main steps including signature gene identification and scoring method (Supplementary Fig. 1). We first performed differential gene expression analysis using one-sided *t* test between TERTp mutant and *ATRX* altered groups. We calculated fold change (FC) using the mean expression of all genes for two groups. The upregulated genes in TERTp mutant group were shortlisted using *P* value threshold ≤0.05 and log 2 FC >1.5.

Using Pearson's correlation, we identified *TERT* co-expressed genes using 81 TERTp mutant cases at thresholds ranging from 0.2 to 0.7 divided by the step of 0.5. In total, it led to 11 successive possible thresholds. For each threshold, we identified a set of co-expressed genes. As the threshold increases, the number of co-expressed genes decreases. We then evaluated how much each threshold elevation affected the gene size change by calculating the percentage difference among the number of genes for two consecutive correlation steps. We used the distribution of this percentage difference to identify a robust threshold. The percentage difference increased with the decreasing number of genes until a drop was observed at 0.6. This suggests that a further increase of the threshold would not remove candidates as effectively as previous thresholds. We thus used the threshold 0.6. This led to the identification of 108 *TERT* co-expressed genes.

The 108 *TERT* co-expressed genes were intersected with upregulated genes identified by differential analysis earlier. The intersected set resulted in 12 genes. Since TERC being an RNA component of telomerase also plays an important role in its function, we added *TERC* to the signature. Pathway enrichment analysis of these genes was done by MSigDB[3].

**Two-step scoring**. We used a two-step approach to score the signature. We first segregated the signature into two components, a constituent component consisting of *TERT* and *TERC* and a marker component consisting of the remaining 11 genes. Let $r_{gi}$ denote the rank of gene *g* in sample *i*, a vector $V_{const}$ is constructed as follows:

$$V_{const,i} = \max\left(r_{tert,i}, r_{terc,i}\right). \tag{1}$$

$V_{marker}$ is calculated based on the 11 marker genes as

$$V_{marker,i} = \sum_{m=1}^{11} r_{m,i}. \tag{2}$$

EXTEND score is calculated as

$$ES_i = \left(\delta V_{const,i} + V_{marker,i}\right) / \left(N_g^* N_m\right), \tag{3}$$

where ES is EXTEND score and $\delta = (1 - corr(V_{const}, V_{marker}))^{-1} \cdot N_g$ and $N_m$ denote the total number of genes and the number of signature genes present in the input data. Finally, scores are linearly scaled to [0,1] for further comparisons.

Thus, for each dataset, the adjusting factor $\delta$ is calculated once reflecting the correlation (Spearman's) between the marker and constituent components. The reason for having this factor is that the size of the constituent component is much smaller than the marker component, thus conceivably its contribution to the final score is also smaller. To balance them, we reason that if the scores of the two components are relatively similar in reflecting the trend, then the constituent component should contribute more to the score since they are functionally deterministic of telomerase activity. Otherwise, a poor correlation suggests that other factors such as *TERT* splicing may affect telomerase activity, thus making *TERT* or *TERC* expression less reliable in predicting telomerase activity. In this scenario, we downplay the contribution of the constituent component by lowering its weight.

In practice, when tested using CCLE cell lines, we found the unadjusted rank-sum scores highly consistent with scores derived from single sample GSEA (Rho = 0.96, *P* value <2.2e − 16), but the computational speed is much faster. *TERT* and *TERC* generally contribute <20% of the final score despite adjustment (Supplementary Fig. 4), emphasizing the significance of the marker component.

Because EXTEND uses rank-based scores, it is insensitive to expression units and scaling normalization methods. One concern is that different gene model annotations used in RNAseq alignment may result in expression matrices of different sizes. For instance, RefSeq contains roughly 20 K genes, whereas GENCODE contains 50–60 K genes. This size difference could alter gene ranks in the expression profiles and thus impact EXTEND scores. However, in our testing, we did not find it concerning. For instance, TCGA used RefSeq as its gene model for RNAseq preprocessing, and the expression matrix contains 20,501 genes, whereas CCLE and GTEx data both used GENCODE and their expression matrices contain >55,000 genes. In either case, we obtained satisfactory results in our benchmark experiments.

**Contribution of Signature Genes towards EXTEND score**. To evaluate the contribution of each signature gene towards the final EXTEND score, we used data from CCLE (downloaded from CCLE website, file CCLE_RNAseq_rsem_genes_tpm_20180929.txt.gz, as of Jan 2019). Because the score was the additive sum of the signature genes (after weight adjustment), we were able to attribute the score to each signature gene. The cell line data show that as the score increases, the contribution from *TERT*/*TERC* also slightly increases, but its contribution rarely

exceeds 20%. In addition, across ~1000 cell lines, the variation of *TERT/TERC* contribution is only ~5%, from 15 to 20%.

**Robustness of EXTEND on sequencing protocol**. We compared EXTEND between poly(A)=enriched mRNA-sequencing protocol and ribosomal RNA depletion protocol using data from GSE51783[50].

**Validation of EXTEND**. Glioma cell line and lung cancer cell line telomerase activity were measured in-house using a modified TRAP and droplet digital TRAP, respectively (see below). Enzymatic activity of BLCA cell lines was requested from Borah et al.[21]. Liposarcoma data were downloaded from GSE14533[52] and neuroblastoma data were obtained from ref. [41]. Only bladder and lung cancer lines included in CCLE were used in our analysis. Glioma expression data were obtained from ref. [84]. In all comparisons of experimentally determined telomerase activity and EXTEND score, we used Spearman's rank test. The differential patterns for neuroblastoma and liposarcomas were tested using two-sided *t* test.

**Telomerase activity in non-neoplastic samples**. Expression data of GTEx was downloaded as of March 2019 (release 2016-01-15_v7). We tested EXTEND scores using both RPKM (reads per kilobase of transcript, per million mapped reads) and count-based expression matrices and the results were highly consistent (Rho = 0.95; *P* value <2.2e − 16). Embryonic tissue data were downloaded from Array Express (accession no: E-MTAB-6814). For each age group, the average was calculated for the curves shown in Fig. 2. GSE81507 was used to calculate telomerase activity in a dyskeratosis congenita case.

**Telomerase activity in tumors**. Expression (in RSEM (RNA-Seq by Expectation-Maximization)) and clinical data of 31 cancer types were downloaded from TCGA PanCan Atlas (https://gdc.cancer.gov/node/905/), including ACC, BLCA, BRCA, CESC (cervical carcinoma and endocervical adenocarcinoma), CHOL, CRC, GBM, HNSC (head and neck carcinoma), ESCA, KICH, KIRC, KIRP, LGG, LIHC (liver carcinoma), LUAD, LUSC (lung squamous cell carcinoma), DLBC (diffuse large B cell lymphoma), OV (ovarian adenocarcinoma), PAAD, PCPG, PRAD, LAML (acute myeloid leukemia), SARC, SKCM (skin cutaneous melanoma), STAD, TGCTs, THYM, THCA, UCEC, UVM (uveal melanoma), and UCS. *TERT* expression status of these tumors was downloaded from ref. [37]. We compared EXTEND scores using two-sided *t* test across tumor and normal samples in 16 cancer types. We next compared EXTEND scores across different tumor stages using two-sided *t* test. Only cancer types that had a minimum of ten cases in each stage were used in this analysis to ensure proper sample size.

For survival analysis, EXTEND scores were categorized into low and high groups based on their median values across each cancer type. Hazard ratios were calculated using an univariate Cox model. Multivariate Cox model was used to control for age and tumor stage.

The ten oncogenic signaling pathways were curated by Sanchez-Vega et al.[69]. We used three to five key genes from each pathway based on frequent alterations across various cancer types. *P* values were subjected to multiple testing using Bonferroni correction.

Tumor subtypes were also curated by Pan-Cancer Atlas. We used Spearman's rank method and Bonferroni correction to correlate EXTEND scores to proliferation, stemness, and whole-genome sequence (WGS)- or whole-exome sequence (WXS)-based telomere lengths. Telomere length data (WGS and WXS) were retrieved from an earlier study[37]. Stemness index, containing gene weights, for TCGA was downloaded from an earlier PanCan study[77]. Accordingly, stemness scores were calculated by correlating the gene weights vector with the gene expression vector per sample using Spearman's correlation.

We performed 1000 permutations for each cancer type by randomly shuffling gene labels of the input expression matrix (RSEM). We then calculated EXTEND scores and stemness scores using the permuted data. Their correlation was compared with the correlation derived from the real data. The empirical *P* values were calculated as the percentage of how many times the permutation data yielded higher correlations than the real data.

*TERT* expression and EXTEND scores were correlated with *POT1* gene across 32 cancer types using Spearman's correlation. Expression data and EXTEND scores were *z*-score transformed across each cancer type in order to reduce tissue effect.

**Single-cell data**. Single-cell data were downloaded from three published studies[78–80]. We calculated the stemness score for the single-cell datasets as mentioned in the section above. We set EXTEND score 0.5 as our criterion to select high telomerase, high stemness cells based on their overall distribution. Differential gene expression analysis was performed using edgeR version 3.27.4. Differentially expressed genes (FDR < 0.01) were subjected to enrichment analysis using MSigDB.

Next, we divided the cells into cycling (G1–S and G2–M phase) and non-cycling cells using the cell cycle signature provided in one of the single-cell studies[78]. The three categories (G1–S, G2–M, and non-cycling) were retrieved using the clustering technique (*K*-means in complex Heatmap R package) based on signature genes. Ambiguous groups (for cycling phases and non-cycling cases) were excluded from

further analysis. Two-sided *t* test was used to compare EXTEND scores of these three groups.

**TRAP assay for glioma cell lines**. The GSC lines, established by isolating neurosphere-forming cells from fresh surgical specimens of human GBM tissue that had been obtained from MD Anderson from 2005 through 2008, were cultured in DMEM/F12 (Dulbecco' modified Eagle's medium/nutrient mixture F12) medium containing B27 supplement (Invitrogen, Grand Island, NY), basic fibroblast growth factor, and epidermal growth factor (20 ng/mL each). Cells were authenticated by testing short tandem repeats using the Applied Biosystems AmpFISTR Identifier Kit (Foster City, CA). The telomerase activity in GSC was determined using TeloTAGGG Telomerase PCR ELISA Kit (Cat# 12013789001, Millipore Sigma) according to the manufacturer's instructions. Briefly, telomerase extract was prepared from 0.2 million GSCs. Cellular telomerase extracts were then used to conduct the TRAP reaction, adding telomeric repeats (TTAGGG) to the 3′ end of the biotin-labeled synthetic primer. The elongation product was then amplified by PCR. The resulting quantity of PCR products depends on the telomerase activity in the cell extracts. A parallel PCR tube containing heated extract for each cell line was included for blank. The PCR products were then denatured and hybridized to a digoxigenin-(DIG)-labeled, telomeric repeat-specific detection probe, and detected using an antibody against DIG (anti-DIG-POD) (concentration: 10 mU/ml). The signal intensity of the antibody was then measured by reading the absorbance of samples at 450 nM with an ELISA plate reader (BMG LABTECH). Relative telomerase activities were calculated using the formula provided by the manufacturer's instructions.

**Digital droplet TRAP assay for non-small lung cancer cell lines**. The data from the non-small cell lung cancer cell lines were previously published[31]. Briefly, one million cells were lysed in NP-40-based telomerase lysis buffer[24] and then diluted in lysis buffer to a cell equivalent concentration of 1250 cells per microliter. One microliter of diluted lysate was added to a telomerase substrate extension assay (final volume of 50 μL and 25 cell equivalents per μL) and incubated at 25 °C for 40 min. Following the extension reaction, each sample (2 μL of extension reaction) was assayed in a droplet digital PCR in triplicate. Thresholds were set on Quantalife (Bio-Rad QX200) software according to previously published methods[31] and telomerase activity is displayed as telomerase extension products per cell equivalents added (50 cell equivalents).

**Reporting summary**. Further information on research design is available in the Nature Research Reporting Summary linked to this article.

## Data availability

Source data generated using TRAP assay is available at this link https://github.com/NNoureen/EXTEND_datacodes. TCGA data were downloaded from Pan-Cancer Atlas (https://gdc.cancer.gov/node/905/; synapse ID: syn4874822). *TERT* promoter mutation status and *ATRX* mutation status were downloaded from our earlier study[37]. CCLE data were downloaded from its website (https://portals.broadinstitute.org/ccle, version 02-Jan-2019). TERC dataset, Liposarcomas, and Neuroblastomas data were downloaded from Gene Expression Omnibus (GEO) (https://www.ncbi.nlm.nih.gov/geo/) ("GSE81507," "GSE14533," and "GSE120572"). GTEx data was downloaded from Genotype Tissue Expression (GTEx) project portal (https://gtexportal.org/home/, v7). Single-cell datasets were downloaded GEO (medulloblastoma: "GSE119926," glioblastoma: "GSE131928," and HNSC: "GSE103322"). Human development dataset was downloaded from Array Express ("E-MTAB-6814"). Source data are provided with this paper.

## Code availability

EXTEND is available as an R package at https://github.com/NNoureen/EXTEND[85]. Analysis codes are available at https://github.com/NNoureen/EXTEND_datacodes.

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

## Acknowledgements

This work was supported by CPRIT (RR170055 to S.Z.) and MD Anderson Cancer Center Brain Cancer SPORE Program Career Enhancement Award (S.Z.). We thank Dr. Zhu Wang (UT Health San Antonio) for discussions on statistical issues and Karen Klein for editorial assistance. The results shown here are in whole or part based on data generated by the TCGA Research Network. The Genotype Tissue Expression (GTEx) Project was supported by the Common Fund of the Office of the Director of the National Institutes of Health, and by NCI, NHGRI, NHLBI, NIDA, NIMH, and NINDS. The data used for the analyses described in this manuscript were obtained from the GTEx Portal.

## Author contributions

S.Z. conceived the study and supervised data analysis. S.Z. and N.N. designed the algorithm. N.N., Y.L., J.Y., J.G., and X.W. performed computational analysis. S.W., W.K. A.Y., and D.K. performed GSC telomerase assays and A.L. performed lung cancer cell line telomerase assays. S.Z. and N.N. wrote the manuscript with input from all authors. The manuscript is approved by all authors.

## Competing interests

The authors declare no competing interests.
