## [Peer Review File · Nature Communications]

Reviewers' Comments:

Reviewer #1:

Remarks to the Author:

Estimation of telomerase enzymatic activity is an important issue in cancer biology. Previous studies mostly used TERT expression to estimate telomerase enzymatic activity, which remains debated. In this study, Noreen et al. developed a simple computational method named EXTEND to predict telomerase enzymatic activity using the expression of a 13-gene signature. This is an interesting study. However, I have some concerns about the rationale of the computational method and the validation of the method.

1. The authors used LGG data in TCGA to identify the 13 genes, the expression of which was used to estimate telomerase activity. They used ALT tumors as controls. Why only used LGG data? This is because the number of TERT tumors and the number of ALT tumors are comparable in LGG. For other types of cancers, the numbers of TERT tumors are much higher than the numbers of ALT tumors. I wonder whether the choice of the 13 genes is LGG specific. That is, the authors should examine whether the genes whose expression used to predict telomerase activity are similar among different types of cancers. In a previous study (Systematic analysis of telomere length and somatic alterations in 31 cancer types, doi:10.1038/ng.3781), TERT expression was reported to be associated with ~73% tumors, whereas ALT only accounts for 5%. Why not use the remaining 22% as controls.
2. The authors compared the performance of EXTEND predictions with those estimated from only TERT expression in some cancer cell lines. EXTEND performed better in most cases. However, in the remaining parts of the manuscript, the authors did not directly compare EXTEND predictions with those estimated from only TERT expression in cancer samples, and non-neoplastic samples. The validation in the present form is not sufficient.
3. The authors should add experiments to knockout some of the 13 genes to examine whether they affect telomerase enzymatic activity actually.

Reviewer #2:

Remarks to the Author:

The manuscript „Inferring Telomerase Enzymatic Activity from Expression Data“ describes and characterizes a gene signature that predicts telomerase activity more accurately than the expression of the telomerase core elements TERT and TERC alone. The EXTEND scoring method is derived from comparison of ALT positive and TERT promoter mutated low grade glioma data, and then its ability to predict telomerase activity is tested on cell lines that were profiled with telomerase activity assays. In the next phase the score is computed for a large number of cancer transcriptomes and compared to other features such as cancer stemness.

The method is made available as R package on an open repository

The work is highly relevant and includes all necessary parts to validate the ability of EXTEND to function as a superior form of telomerase activity assay surrogate when compared to TERT expression. Some aspects regarding the method itself are not documented optimally. As the method relies on gene ranks it is most important to document which genes are included into this ranking. An example computation which reports how the score is computed for an individual case may help to clarify this point. Some of the issues below are clarified after reading the supplementary figures, but should already be addressed in the text.

Major comments:

- Multiple of the genes included into the signature are directly related to telomeres or cancer, while others have more obscure or unknown functions. Therefore, the genes that belong to the signature should be briefly discussed in the corresponding section of the manuscript.
- Include the tables with the t-tests and the co-expression analysis that were used to identify the EXTEND gene signature as supplementary data. Order these table by p-value, respectively correlation

coefficient, and mark the genes you selected for further analysis.

Minor comments on Material and methods:

- Add the names of the files you used for each analyses
- Add the unit in which the gene expression was quantified in (e.g. TPM, FPKM etc.)
- Line 15-16: state what varying "thresholds from 0.2 to 0.7" refers to. If these are correlation coefficients state with which method it was computed (e.g. Pearson)
- How are the ranks of the signature genes computed exactly? What is the input value: gene expression, normalized gene expression, differential gene expression? Is in all datasets the number of genes in you compute the rank on the same? What gene model is used (GENCODE, ENCODE, RefSeq)? This information is relevant as readers who try to reproduce your work using a different gene model may end up with different rankings.
- If possible present a formula that represents the scoring schema.
- Line 131-133: The second sentences appears to be incomplete.
- Is the scaling factor computed once, or is it recomputed for each new dataset?
- Supplementary Table 1: This table is not easy to read as it is truncated and placed on two pages. This should

Reviewer 1 comments

Estimation of telomerase enzymatic activity is an important issue in cancer biology. Previous studies mostly used TERT expression to estimate telomerase enzymatic activity, which remains debated. In this study, Noureen et al. developed a simple computational method named EXTEND to predict telomerase enzymatic activity using the expression of a 13-gene signature. This is an interesting study. However, I have some concerns about the rationale of the computational method and the validation of the method.

We thank the reviewer for the positive comments.

1. The authors used LGG data in TCGA to identify the 13 genes, the expression of which was used to estimate telomerase activity. They used ALT tumors as controls. Why only used LGG data? This is because the number of TERT tumors and the number of ALT tumors are comparable in LGG. For other types of cancers, the numbers of TERT tumors are much higher than the numbers of ALT tumors. I wonder whether the choice of the 13 genes is LGG specific. That is, the authors should examine whether the genes whose expression used to predict telomerase activity are similar among different types of cancers. In a previous study (Systematic analysis of telomere length and somatic alterations in 31 cancer types, doi:10.1038/ng.3781), TERT expression was reported to be associated with ~73% tumors, whereas ALT only accounts for 5%. Why not use the remaining 22% as controls?

We thank the reviewer for raising this great question. The reason we chose LGG as our training cohort is that it has a large sample size for both TERT promoter mutant tumors and ATRX mutant tumors (ALT) (Supplementary Fig 2A), as the reviewer rightfully pointed out. Importantly, these two genotypic groups demonstrate strong mutual exclusivity suggesting they employ two distinct telomere maintenance mechanisms (Supplementary Fig 2B). While positive telomerase is commonplace in cancer, the only other tumor type in TCGA that has relatively abundant ALT cases is sarcoma (SARC), but the case number is nowhere near LGG (Supplementary Fig 2A). Moreover, SARC is a heterogeneous group consisting of many disease entities such as synovial sarcoma, liposarcoma, myxofibrosarcoma etc. (PMID: 29100075). This lineage heterogeneity may significantly confound signature identification. We explained our rationale for using LGG in the first paragraph of the result section.

The issue with using the 22% double wildtype (wt/wt) tumors (no detectable TERT expression, no ATRX/DAXX mutation) is that this group is likely an admixture of telomerase-driven and ALT-driven telomere maintenance mechanisms. In this study by us (Barthel et al. 2017), we found this wt/wt group has average longer telomeres in some cancer types but shorter telomeres in others. The absence of TERT expression in these tumors is not a reliable indication of its telomere maintenance mechanism because TERT is located in a region with high GC content, which negatively affects sequencing. In addition, these wt/wt tumors disproportionately come from indolent cancer types. Thus, without further experimental or other evidence, we felt there are too many confounding factors to control if using them as training set.

To address the reviewer's question whether this 13-gene signature is LGG specific, we correlated the expression of the 11 marker genes (excluding TERT and TERC) with TERT across TCGA cohorts. The result is shown below in Figure R1. In the figure, positive correlation is shown in red, and the bars on the right indicate how many of the genes pass p value 0.05 (the same criterion applied to LGG in signature identification). In the majority of cancer types, most markers show positive correlation with TERT, suggesting this signature is not specific to LGG. Across pan-cancer, all genes show positive correlation. However, we do observe in rare indolent cancer types such as cholangiocarcinoma (CHOL), kidney

chromophobe (KICH) and Pheochromocytoma and Paraganglioma (PCPG), the correlations seem to be worse. The patterns for CHOL and KICH do not appear to be very distinct from other cancer types, thus the relatively smaller number of genes passing the threshold is likely due to their small sample sizes. PCPG has the highest incidence of double wildtype tumors (88%) in TCGA (see Fig 5A in Barthel et al. 2017). The absence of TERT expression values likely contribute to this worse correlation. Nevertheless, the result clearly indicates that this signature is cancer type agnostic.

We also gently remind the reviewer that cell lines used in our validation are from multiple cancer types including brain, lung, and bladder. In all three, EXTEND can predict telomerase activity thus substantiating its use in broad lineage contexts.

In the revision, we added comments in the discussion to point out the possible limitation of EXTEND in indolent cancer types such as PCPG (second paragraph in Discussion). We also added Figure R1 as supplementary Figure 3 and added comments in the corresponding text (second paragraph in Results). We thank the reviewer for this insightful comment that has resulted in a strengthened manuscript.

Figure R1. Correlation of TERT expression with 11 marker genes of the EXTEND signature across 32 cancer types in TCGA. Color intensity of heatmap represents correlation coefficient, while the bar plot on right hand side represents number of genes passing p-value threshold of 0.05. The numbers on the left of the heatmap indicate sample size.

2. The authors compared the performance of EXTEND predictions with those estimated from only TERT expression in some cancer cell lines. EXTEND performed better in most cases. However, in the remaining parts of the manuscript, the authors did not directly compare EXTEND predictions with those estimated

from only TERT expression in cancer samples, and non-neoplastic samples. The validation in the present form is not sufficient.

We thank the reviewer for acknowledging our cell line validation results. We did compare TERT and EXTEND in non-neoplastic sample analysis, and we apologize for omitting it in cancer sample analysis. In the revision, we have added them in the revised manuscript, see below.

For non-neoplastic samples, we included both EXTEND scores and TERT expression in our results (Figures 2A-C). We discussed extensively in GTEx data analysis how EXTEND scores differ from TERT expression using skin transformed fibroblasts, brain and testis tissues (see Figure 2A and line 158-164 in the initial submission). To further illustrate this, below is our Figure 2B showing EXTEND score and TERT expression of the heart tissue from prenatal stage to adulthood. EXTEND scores show a drop between 12th and 13th embryonic week, a pattern consistent with findings reported in previous publications (Cardoso-Moreira, M. et al.2019; Ulaner & Giudice, 1997), whereas TERT expression is not informative because it is hardly detected after 7 weeks.

Figure 2B from the manuscript. EXTEND scores and TERT expression across human embryonic heart development. The left y-axis indicates TERT expression (blue) while the right y-axis represents EXTEND score (red). The lines are regressed by averaging samples from each age group.

We used two cancer patient sample datasets, liposarcoma and neuroblastoma, for validation. The idea was that these cancer types have high incidences of ALTs thus the contrast in telomerase activity between ALT and telomerase positive tumors provides a basis for validation. The neuroblastoma data was obtained from Ackerman et al. science, 2018 (PMID 30523111). As we explained in the manuscript, neuroblastomas can be divided into tumors without telomere maintenance mechanism (TMM), ALT, and positive telomerase. The latter is further split into MYCN amplified, TERT expression high, and TERT rearrangement. The Ackerman study compared TERT expression and telomerase activity in 52 tumors, see panel A Figure R2. As expected, telomerase positive tumors have higher telomerase than ALT and tumors without TMM ($p < 2.2e-16$). Interestingly, the TERT high group has a lower median telomerase activity than MYCN amplification group despite a higher average TERT expression. In our analysis, EXTEND scores recapitulate this discrepancy (Figure R2, panel B) showing lower scores for the TERT high group than MYCN amplified tumors. We note that the difference in telomerase activity between the two groups in the Ackerman study did not reach statistical significance, likely due to very small sample sizes. However, the consistency between EXTEND and experimentally determined telomerase activity

further substantiate our observations from cell lines that EXTEND is superior to TERT expression in predicting this important feature of the telomerase.

In Figure R3, we provide side by side comparison between EXTEND and TERT expression for the liposarcoma dataset (Lafferty-Whyte, K. et al. PMID 19684619). The dataset consists of both cell lines and tumor samples. We conducted a pairwise comparison using t test. We found that both TERT expression and EXTEND separate ALT from telomerase samples, but EXTEND achieves a better separation (p value 7e-6 vs 0.003 for cell lines, 0.001 vs 0.009 for tumor samples). The reason is that variance is much higher for TERT expression than EXTEND scores. We emphasize that this dataset is not ideal to compare TERT and EXTEND because tumors were only broadly divided into ALT and telomerase

positive without quantitative telomerase activity. It is conceivable that both TERT and EXTEND can separate these two groups.

We have added both Figure R2 and R3 to the revised manuscript as supplementary Figures S9 and S10.

3. The authors should add experiments to knockout some of the 13 genes to examined whether they affect telomerase enzymatic activity actually.

The 13-gene signature is collectively a marker of telomerase activity, as the reviewer is aware. In scoring the signature, we separate the signature into a constituent component (TERT and TERC) and a marker component (the remaining 11 genes) (line 110 in the first submission). We never claimed in our manuscript that any of the marker genes is functionally involved in regulating telomerase activity. In our discussion (line 308-309 in the initial submission), we reiterated that these marker genes “have no reported functional associations with the telomerase.” Thus, we respectfully argue that this suggestion is out of the scope of this study.

Reviewer 2 comments

The manuscript “Inferring Telomerase Enzymatic Activity from Expression Data” describes and characterizes a gene signature that predicts telomerase activity more accurately than the expression of the telomerase core elements TERT and TERC alone. The EXTEND scoring method is derived from comparison of ALT positive and TERT promoter mutated low grade glioma data, and then its ability to predict telomerase activity is tested on cell lines that were profiled with telomerase activity assays. In the next phase the score is computed for a large number of cancer transcriptomes and compared to other features such as cancer stemness.

The method is made available as R package on an open repository. The work is highly relevant and includes all necessary parts to validate the ability of EXTEND to function as a superior form of telomerase activity assay surrogate when compared to TERT expression. Some aspects regarding the method itself are not documented optimally. As the method relies on gene ranks it is most important to document which genes are included into this ranking. An example computation which reports how the score is computed for an individual case may help to clarify this point. Some of the issues below are clarified after reading the supplementary figures but should already be addressed in the text.

We thank the reviewer for acknowledging the significance of this work and our validation efforts, and we apologize for the lack of clarity in describing methods and data. In the revision, we have added more details, including mathematical description of the method, in Methods, and also added additional data per the reviewer’s suggestion. In addition, we added a diagram in Supplementary Fig. 1 to graphically illustrate the computing procedure. We think these changes improve clarity and readability. We thank the reviewer for prompting us to make these helpful changes.

Major comments:

1. Multiple of the genes included into the signature are directly related to telomeres or cancer, while others have more obscure or unknown functions. Therefore, the genes that belong to the signature should be briefly discussed in the corresponding section of the manuscript.

We have the following discussions on the signature genes. Interestingly, none of the signature genes except TERT seem to be well recognized cancer genes. Underlined texts were added to the manuscript in the revision.

“Seven of the 13 genes were highly expressed in testis but low in other tissues. None of the signature genes except TERT was catalogued by the expert curated Cancer Gene Census (as of July 2020), though LIN9 and HELLS were recently implicated in cancer (PMID: 32054769 and PMID: 31541170). This suggests the signature is largely not cancer specific. Mutations in HELLS, a gene encoding a lymphoid-specific helicase, cause the centromeric instability and facial anomalies (ICF) syndrome, a genetic disorder associated with short telomeres⁴¹. Another signature gene POLE2 was a subunit of DNA polymerase epsilon, a complex previously linked to telomerase c-strand synthesis (PMID: 26883631). A summary of the signature genes, including their tissue expression pattern, function, and expression pattern in LGG, was provided in **Supplementary Table 1**. Pathway enrichment analysis suggested an overrepresentation of the signature genes in cell cycle (FDR=1.95e-4), particularly S phase (FDR=0.01, **Supplementary Table 2**), a narrow time window when telomerase is active in extending telomeres⁴².”

2. Include the tables with the t-tests and the co-expression analysis that were used to identify the EXTEND gene signature as supplementary data. Order these table by p-value, respectively correlation coefficient, and mark the genes you selected for further analysis.

We screened all genes in the TCGA expression matrix (n=20501, downloaded from PanCan Atlas) to identify the signature genes. In the revision, we have included the 108 TERT co-expressed genes and their P-values, correlation coefficients and log2FC between TERT promoter mutants and ALT (ATRX mutants) in supplementary Table 1 with the signature genes highlighted.

Minor comments on Material and methods:

1. Add the names of the files you used for each analyses

Done.

2. Add the unit in which the gene expression was quantified in (e.g. TPM, FPKM etc.)

Done.

3. Line 15-16: state what varying “thresholds from 0.2 to 0.7” refers to. If these are correlation coefficients state with which method, it was computed (e.g. Pearson)

Done.

4. How are the ranks of the signature genes computed exactly? What is the input value: gene expression, normalized gene expression, differential gene expression? Is in all datasets the number of genes in you compute the rank on the same? What gene model is used (GENCODE, ENCODE, RefSeq)? This information is relevant as readers who try to reproduce your work using a different gene model may end up with different rankings.

The algorithm starts with ranking genes based on their expression values in each sample (see updated method). Because this is ranking, it is insensitive to expression units or normalization procedures applied to the input dataset. As the reviewer pointed out, the size of the input dataset, i.e. the number of genes, impacts the ranks. In the algorithm, we normalize scores by the total number of genes of the input dataset. At times, not all signature genes are present in an input dataset, we thus also normalize the sum by the size of the presented signature in the dataset. This process relies entirely on the input data matrix. Certainly, if a different gene model system is used in RNAseq alignment, the expression matrix (number of genes, expression values etc.) will change so is the rank sum score. However, we don't find this is a significant issue as we have used data derived from various gene models in our testing. For instance, TCGA used RefSeq for its RNAseq alignment (20,501 genes), whereas GTEx and CCLE both used GENCODE (both have >55,000 genes in expression data), but we did not see much difference in algorithm performance. In the initial phase of the algorithm development, we also tested gene filtering such as removing lowly expressed genes etc., the results virtually remained the same.

We have added these discussions to Methods.

5. If possible present a formula that represents the scoring schema.
Done.
6. Line 131-133: The second sentences appears to be incomplete.
Revised.
7. Is the scaling factor computed once, or is it recomputed for each new dataset?
It is computed for each new dataset.
8. Supplementary Table 1: This table is not easy to read as it is truncated and placed on two pages.
We have adjusted the Supplementary Table 1.

Reviewers' Comments:

Reviewer #1:

Remarks to the Author:

In the revised manuscript, the authors have significantly improved the manuscript and addressed my concerns.

Reviewer #2:

Remarks to the Author:

The revision significantly improved the manuscript and all of my comments have been addressed.

Regarding the mathematical description of the model in lines 394-399 I remain with a last comment: You switched to a vector calculation in line 396, which is a bit inconsistent with the lines 394 and 395. Changing the line 396 as follows ('_' should denote subscript): $ES_i = (\text{delta } V_{\text{const},i} + V_{\text{marker},i}) / (N_g * N_m)$, would make it clear that ES is computed for each patient i individually, in contrast to the computation of the scaling factor delta in the next line which is derived on the whole dataset.

Reviewer #1 (Remarks to the Author):

In the revised manuscript, the authors have significantly improved the manuscript and addressed my concerns.

We are grateful to Reviewer #1 for taking time to evaluate our work.

Reviewer #2 (Remarks to the Author):

The revision significantly improved the manuscript and all of my comments have been addressed.

Regarding the mathematical description of the model in lines 394-399 I remain with a last comment:

You switched to a vector calculation in line 396, which is a bit inconsistent with the lines 394 and 395. Changing the line 396 as follows ('_' should denote subscript): $ES_i = (\Delta V_{const,i} + V_{marker,i}) / (N_g * N_m)$, would make it clear that ES is computed for each patient i individually, in contrast to the computation of the scaling factor Δ in the next line which is derived on the whole dataset.

We have made the revisions suggested by the reviewer.